# Do LLMs Signal When They're Right? Evidence from Neuron Agreement

**Kang Chen** [* 1]  **Yaoning Wang** [* 1]  **Kai Xiong** [1]  **Zhuoka Feng** [1]  **Minshen Yu** [1]  **Wenhe Sun** [1]  **Haotian Chen** [1]
**Yixin Cao** [1 2]

## Abstract

Large language models (LLMs) commonly boost reasoning via sample-evaluate-ensemble decoders, achieving label free gains without ground truth. However, prevailing strategies score candidates using only external outputs such as token probabilities, entropies, or self evaluations, and these signals can be poorly calibrated after post training. We instead analyze internal behavior based on neuron activations and uncover three findings: (1) external signals are low dimensional projections of richer internal dynamics; (2) correct responses activate substantially fewer unique neurons than incorrect ones throughout generation; and (3) activations from correct responses exhibit stronger cross sample agreement, whereas incorrect ones diverge. Motivated by these observations, we propose Neuron Agreement Decoding (NAD), an unsupervised best-of-N method that selects candidates using activation sparsity and cross sample neuron agreement, operating solely on internal signals and without requiring comparable textual outputs. NAD enables early correctness prediction within the first 32 generated tokens and supports aggressive early stopping. Experiments demonstrates the effectiveness and efficiency of our analysis and the proposed method.

## 1. Introduction

Large language models (LLMs) have demonstrated remarkable reasoning capabilities (Wei et al., 2022; Tang et al., 2024). To further boost their performance, some methods leverage answer consistency by selecting the best response from multiple samples, marked as sample-evaluate-ensemble. Notably, this approach requires no ground truth

labels, essentially providing a "free lunch" improvement. Beyond inference-time applications, this paradigm has also been integrated into unsupervised reinforcement learning, enabling models to train at larger scales without ground truth, thereby raising the performance ceiling.

Clearly, the core challenge of this paradigm lies in how to evaluate the quality of multiple samples. The most straightforward approach is majority voting (Wang et al., 2022); (Chen et al., 2023) prompts the model to self-evaluate before ensemble and select the most consistent answer among candidates. Another line of work explores the model's response confidence, leveraging output states from the forward pass (e.g., token probabilities) to evaluate and ensemble responses. Typically, high-confidence responses exhibit better quality and low-confidence ones shall be pruned (Fu et al., 2025). However, these methods rely solely on model outputs, with confidence or entropy metrics based on token probabilities — what we define as the external behaviors of LLMs. This raises critical questions: Do models inherently encode response quality signals in their outputs? How reliable are such assessments? According to GPT-4 reports (Achiam et al., 2023), LLMs lose calibration capabilities after post-training, showing no clear linear relationship between token probability and response correctness. This appears to contradict the effectiveness demonstrated in existing work.

In this paper, we further investigate the model's internal behaviors, neuron activation vectors, to explore their relationships with external behaviors and response correctness. Through extensive experiments, we reveal that: 1) External behaviors (e.g., entropy) represent low-dimensional projections of internal behaviors. This is intuitive, as token probabilities determine output tokens and are themselves determined by neuron activations across layers. 2) Consequently, internal behaviors contain richer signals, leading to our discovery of their relationship with response quality — correct responses activate significantly fewer neurons compared to incorrect ones. 3) Finally, through visualization, we observe patterns among correct responses. They tend to activate similar unique neurons. These novel patterns in LLMs' internal behaviors inspire better evaluation methods and more efficient assessment of sampled response correctness, ultimately yielding superior ensemble results.

[1]Fudan University, Shanghai, China [2]Shanghai Innovation Institute, Shanghai, China. Correspondence to: Yixin Cao <yxcao@fudan.edu.cn>.

*Proceedings of the 43rd International Conference on Machine Learning*, Seoul, South Korea. PMLR 306, 2026. Copyright 2026 by the author(s).

Building on these insights, we propose **Neuron-Agreement Decoding (NAD)**, an unsupervised method that selects high-quality reasoning trajectories solely based on internal neuron activations. Specifically, NAD favors either responses with minimal neuron activations or those that exhibit the greatest agreement with other sampled responses. Leveraging such signals also enables us to predict response correctness at a very early stage of generation (e.g., within the first 32 tokens), rather than requiring full sequences as in external ensemble methods, thereby substantially improving the efficiency and effectiveness of sample-then-ensemble decoding.

For evaluation, we consider both tasks with easily verifiable correctness (with ground truth, suitable for majority voting) and more challenging scenarios (without ground truth or with multiple valid solutions, such as code generation), validating our method's effectiveness. Moreover, when combined with an early-stopping strategy, NAD reduces token consumption in parallel sampling by up to two orders of magnitude while delivering superior performance. Our contributions are summarized as follows:

- We conduct a deep investigation into the relationships among internal neuron activation, external behaviors, and response correctness in LLMs.

- We design a Neuron-Agreement Decoding (NAD) method for scalable best-of-N sampling.

- Experimental results validate the effectiveness of our findings and method, while the efficiency gains are also achieved through the early-stopping strategy.

## 2. Related Work

**Outputs-based Voting.** Self-consistency (Wang et al., 2022) is a test-time ensemble technique that samples multiple chain-of-thought (CoT) solutions from an LLM and selects the final answer by majority vote. This method significantly improved performance on arithmetic and commonsense QA benchmarks, revealing that while any single chain might be incorrect, aggregating multiple solutions can correct errors. Variants of this idea include Soft Self-Consistency (Wang et al., 2024), which gives partial credit to similar answers rather than exact match voting. Self-consistency works well when outputs are relatively constrained (e.g., numerical or factual answers), but is less applicable to open-ended generation, where answers cannot be easily compared for voting.

**Confidence-Based Selection.** Another line of work exploits internal confidence or uncertainty metrics. (Kang et al., 2025) introduced self-certainty, which uses the model's token-level probabilities to estimate the confidence

of each reasoning chain. In their best-of-$N$ selection framework, self-certainty scores guided the choice of the final answer, achieving better scaling with $N$. (Fu et al., 2025) proposed DeepConf, which monitors token prediction entropy during generation to prune low-confidence reasoning paths on the fly, thereby saving computation while maintaining accuracy. These approaches require access to the model's probability distribution at each step. However, these approaches still rely on the availability of comparable answers, which limits their applicability.

## 3. Analysis of Internal Behaviors

While prevailing Best-of-N methods predominantly rely on external output signals, they often overlook the rich internal dynamics governing model reasoning. Inspired by recent findings on the mechanism interpretability works (Cao et al., 2025), we design a systematic framework to decode these internal behaviors. In this section, we first operationalize neuron activations as high-dimensional state vectors and describe similarity metrics to quantify their consensus and sparsity (Section 3.1). We then proceed with three progressive analyses to validate our hypotheses: i) the relationship between internal activations and external metrics (Section 3.2), demonstrating that internal signals encode richer structural consensus. ii) We examine the link between activation patterns and model performance, uncovering a sparsity-correctness correlation consistent with the "sharpening" effect of alignment. And, iii) we analyze the temporal evolution of these signals, confirming that correctness is distinguishable at the early prefix stage (Section 3.3).

### 3.1. Definitions and Metrics

We first define the Neuron Activation Set as model internal representation, followed by metrics for analysis.

**Preliminaries.** Neuron activation patterns can reveal the internal dynamics of LLMs, which are closely associated with specific types of abilities in LLMs (Pan et al., 2024; Templeton et al., 2024). Here, we adopt the definition of activated neurons of (Cao et al., 2025). Given an LLM with an input $x$, it can generate an output token sequence $\boldsymbol{y} = (y_1, y_2, \ldots, y_t)$ from a single sampling, the SwiGLU-based (Chowdhery et al., 2023) contribution of neuron $i$ in layer $l$ to output token $y_j$ as:

$$f_{\text{neuron}}(i, l, y_j \mid \boldsymbol{y}_{<j}) = \left(\mathbf{W}_u \mathbf{W}_{\text{out}}^l \circ \text{SiLU}\left(\mathbf{x}_t^l \mathbf{W}_g^l\right)\right)_{y_j, i}, \tag{1}$$

where $\boldsymbol{y}_{<j} = (y_1, y_2, \ldots, y_{j-1})$ denotes the response sequence before the $j$-th token $y_j$, SiLU is the Swish activation (Shazeer, 2020), $\mathbf{W}_{\text{out}}^l, \mathbf{W}_g^l$ are the output/gate projections in FFN, $\mathbf{W}_u$ is the unembedding matrix transforming the hidden states into distributions over the vocabulary, $\circ$ is an element-wise product with broadcasting, and $\mathbf{x}_{j-1}^l$

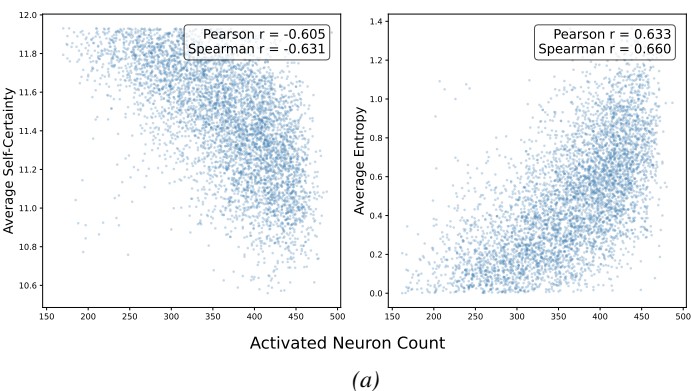
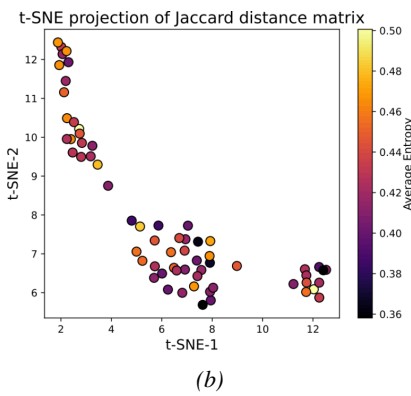

*(a)*                      *(b)*

*Figure 1. (a)* Scatter plots of the number of activated neurons versus confidence-based metrics (Self-Certainty and Entropy). The neuron counts show significant correlations with both metrics, indicating that the activated neuron states provide a high-dimensional representation of traditional confidence measures. For more detailed visualizations that distinguish between correct and incorrect samples, please refer to Figure 10 in the Appendix. *(b)* t-SNE representation of activated neurons, with point colors indicating the average entropy of the corresponding samples. No clear consistency is observed between entropy and the resulting clusters, suggesting that the activated neurons contain high-dimensional structural information not captured by entropy.

denotes the hidden input of token $y_{j-1}$ to the FFN at $l$-th layer. For a given threshold $\eta$, the activated neuron set for a sample $(x, \boldsymbol{y})$ is defined as:

$$N_{\text{activated}}(x, \boldsymbol{y}) = \left\{ (i, l) \mid \exists y_j \in \boldsymbol{y}, \ f_{\text{neuron}}(i, l, y_j \mid x \oplus y_{<j}) > \eta \right\} \quad (2)$$

where $l = 1, 2, ..., L$ represents the layer index, and $i = 1, 2, ..., N$ indicates the neuron index in each layer. The implementation of the threshold function $\eta$ can be found in Appendix B.

**Neuron Activation Set.** In this work, we refine the definition into a more fine-grained form to better capture activations throughout the reasoning process. Specifically, we divide the entire reasoning process into $B$-sized chunks $\boldsymbol{y} = (\boldsymbol{y}_1, \boldsymbol{y}_2, ..., \boldsymbol{y}_{\lceil t/B \rceil})$, compute the activation set for each chunk using the definition in Eq.(2), and then take their union:

$$N_{\text{activated}}(x, \boldsymbol{y}) = \bigcup_{i=1}^{\lceil t/B \rceil} N_{\text{activated}}(x, \boldsymbol{y}_i) \quad (3)$$

This modification allows us to better capture localized information within the reasoning process.

**Analysis Metrics.** To systematically decode the relationship between internal dynamics and scalar metrics of model performance, we introduce two quantitative metrics based on the above definition of neuron activation set.

First, we quantify the volume of the model's active subnetwork by counting the number of activated neurons per reasoning chunk, and then measure the correlation between neuron counts and the conventional scalar metrics, such as self-certainty and entropy. Second, to uncover structure in how different samples activate neurons, we embed samples with t-SNE using the Jaccard index between activated-

neuron sets as the similarity measure, i.e.

$$S_{ij} = \frac{|N_{\text{activated}}(x, \boldsymbol{y}_i) \bigcap N_{\text{activated}}(x, \boldsymbol{y}_j)|}{|N_{\text{activated}}(x, \boldsymbol{y}_i) \bigcup N_{\text{activated}}(x, \boldsymbol{y}_j)|}. \quad (4)$$

Where $\boldsymbol{y}_i, \boldsymbol{y}_j$ are different responses.

### 3.2. Exp I: Internal Signals Encode Richer Information

Concretely, we use `Qwen3-4B-Think` to generate 64 responses per instance on `AIME24`. We hypothesize (operationally) that a model's internal dynamics during inference — captured as the set of activated neurons — can be viewed as giving rise to approximate low-dimensional summaries such as self-certainty; reducing behavior to such scalars discards much of the underlying high-dimensional information.

We test this via two complementary findings: (1) **activated neurons can, to some extent, aggregate into those scalar metrics**, i.e., the set of activated neurons contains the information needed to compute those metrics; and (2) **activated neurons exhibit patterns that scalar metrics cannot capture**. The results, shown in Figure 1a, demonstrate significant correlations (with p-value less than 0.05): neuron count correlates positively with entropy and negatively with self-certainty, thereby supporting (1).

On the other hand, the t-SNE visualization in Figure 1b colors each sample by the sequence's average entropy. The plot reveals clear clustering of responses by their activated-neuron patterns; importantly, samples within the same cluster do not necessarily share similar entropy values, and samples from different clusters can exhibit similar entropy. This indicates that the activated-neuron patterns encode high-dimensional structure that scalar metrics such as entropy cannot represent.

Taken together, these results suggest that confidence-type

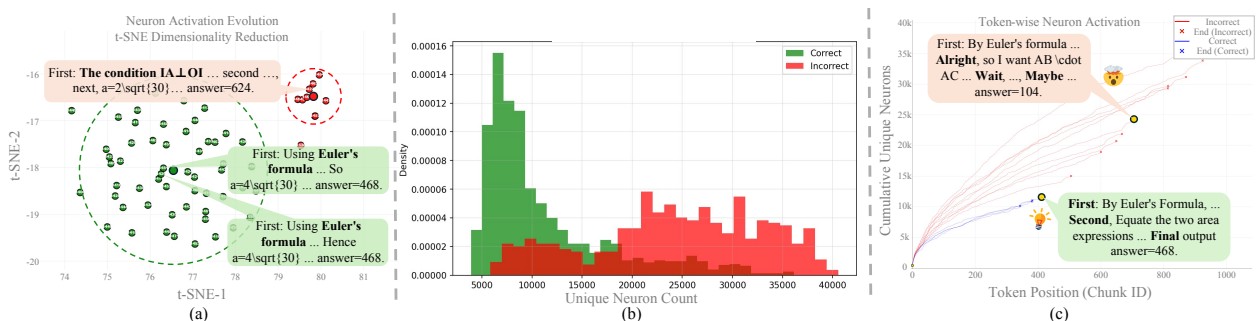

*Figure 2.* Results on AIME24. (a) t-SNE of responses to one prompt: center clusters share similar reasoning; outliers diverge. (b) Correct answers activate far fewer neurons than incorrect ones. (c) Token-wise trajectories show incorrect responses repeatedly shift strategies, engaging more neurons. These observations motivate **Insight 1** and **Insight 2** presented in Section 3.2.

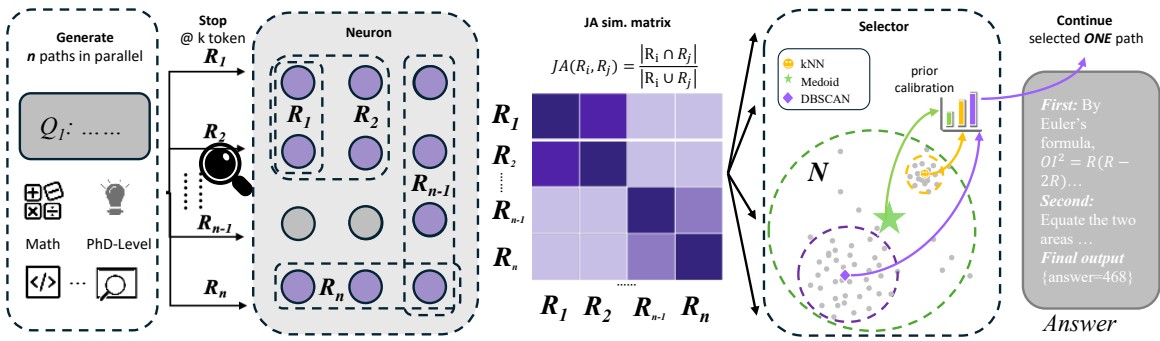

*Figure 3.* Framework of Neuron Agreement Decoding (NAD). NAD selects high-quality answers by leveraging the consensus of internal neuron activations during the sampling process, without relying on canonical textual outputs. This consistency can be identified using the proposed kNN-based approach, among others. Moreover, this procedure can be applied at an early stage of sequence generation, pruning low-quality responses in advance and reducing token usage.

scalar metrics are effectively low-dimensional projections of high-dimensional activated-neuron states[1]. This observation naturally raises the question: if scalar confidence can guide answer selection, can we exploit richer, higher-dimensional state information to guide selection more effectively? To explore this possibility, in the next section we conduct a deeper analysis of the relationship between neuron activation patterns and response correctness.

### 3.3. Exp II & III: Sparsity-Correctness Correlation and Early Emergence

To investigate the correlation between neuron activation patterns and model performance, we continue with the same experimental setup, while recording both the correctness of each response and its corresponding set of activated neurons.

Figure 2(a) shows the visualization of multiple samples for selected instances, where green points correspond to correct responses and red points to incorrect ones. We observe two clear patterns: 1) Neuron activation patterns across different responses tend to form natural clusters, indicating

a form of consensus; 2) Incorrect responses tend to lie at the margins of clusters, farther from neighboring samples, whereas correct reasoning trajectories align more tightly with the central distribution. These lead to our first key insight:

**Insight 1.** By leveraging neuron activation patterns across responses, we can define consensus and identify reasoning trajectories more likely to be correct without requiring text-level matching.

Complementing the clustering result, we compare neuron activations for correct vs. incorrect samples during generation (Fig. 2(b,c)). In Fig. (b), correct samples consistently exhibit substantially fewer activated neurons than incorrect ones, consistent with the view that successful trajectories balance exploration and exploitation—reaching answers with fewer trial-and-error steps—whereas failures over-explore. Fig. (c) tracks the growth of unique activated neurons with generated tokens for each sampling; correct trajectories activate fewer neurons throughout. This pattern aligns with the overall statistics and suggests using activation signals to terminate low-quality generations early. These observations lead to our second key insight:

**Insight 2.** The number of activated neurons in the early

---

[1]We cannot exhaust all possible activated neuron patterns and their mappings to scalar metrics. In future work, we aim to explore more combinations.

stage of generation can serve as a signal to distinguish high-quality answers from low-quality ones. Specifically, high-quality answers tend to activate fewer neurons. Aligned by generation step (Fig. 2(c)), correct chains activate fewer neurons at matched tokens, indicating a non-length effect and motivating chunk-conditioned early stopping.

Building on these two insights, we can design corresponding algorithms that leverage neuron activation patterns across samples to uncover consensus and identify higher-quality answers.

## 4. Methodology

Building on the insights from above investigation, we aim to operationalize two key observations regarding neuron activations in sampled reasoning trajectories. These observations suggest two complementary strategies for selecting high-quality reasoning trajectories. In the following, we introduce two strategies. Figure 3 illustrates the overall framework.

### 4.1. Neuron-Agreement Decoding (NAD)

To operationalize the notion of *consensus*, we build directly on **Insight 1**. Samples of the same input exhibiting similar neuron activation patterns tend to correspond to correct reasoning, while incorrect ones deviate substantially. We capture consensus by exploiting these internal dynamics of the model.

Concretely, for a given input $x$, we generate $n$ sampled trajectories $\{(x, \boldsymbol{y}_i)\}_{i=1}^{n}$, and obtain their activated neuron sets $\{N_{\text{activated}}(x, \boldsymbol{y}_i)\}_{i=1}^{n}$ as defined in Section 3.1. Pairwise similarities among samples, already formalized by the Jaccard index in Section 3.2, provide a relational structure over the sampled responses. The resulting consensus matrix $S \in [0,1]^{n \times n}$ captures agreement among samples at the level of neuron activations.

Building upon this representation, our goal is to identify consensus samples — those most consistent with others in their neuron activations — and thereby select trajectories that are more likely to be correct. We propose the following structure discovery methods:

$k$**NN-Agreement.** For each sample $i$, compute the sum of its top-$k$ pairwise similarity scores to other solutions, denoted as $s_i$. Select the solution $\hat{i}$ with the highest $s_i$.

**Global Medoid.** The medoid denotes the point that minimizes the total distance to all other samples. Under the Jaccard Index metric, this corresponds to maximizing the sum of similarities:

$$\hat{i} = \arg\max_{i} \sum_{j=1}^{n} S_{ij}.$$

**DBSCAN.** We apply the clustering algorithm to the distance matrix $D = 1 - S$ to identify clusters. Select the largest cluster $C$, then find the medoid within this cluster:

$$\hat{i} = \arg\max_{i} \sum_{p \in C} \sum_{q \in C} S_{pq}.$$

Building on **Insight 2**, we propose an alternative selection strategy: since correct reasoning trajectories tend to activate relatively fewer neurons, we choose the trajectory with the fewest activated neurons among the $n$ samples:

$$\hat{i} = \arg\min_{i} |N_{\text{activated}}(x, \boldsymbol{y}_i)|.$$

This approach does not rely on pairwise similarities between samples; instead, it treats the number of activated neurons in a single sample as a proxy for its quality, allowing us to select trajectories that are more likely to be correct efficiently. We denote this approach as **MinAct**.

### 4.2. Integrate NAD with Confidence Mechanism

NAD essentially provides a way to perform majority voting without relying on text matching. This step is orthogonal to confidence-based filtering strategies such as DeepConf (Fu et al., 2025). Therefore, we can first apply a confidence-based method to filter the generated trajectories, and then apply NAD to further improve the quality of the generated sequences. We adopt the tail-token confidence strategy from DeepConf. To be specific, let $P_t(j)$ denote the probability of the $t$-th token in a trajectory generating the $j$-th top-$k$ token. The token confidence score is defined as

$$C_t = -\frac{1}{k} \sum_{j=1}^{k} \log P_t(j) \tag{5}$$

We compute the average confidence score over the last $T_{tail}$ tokens of the generated sequence, filter out the bottom $p\%$ low-confidence sequences, and then apply the NAD procedure described in Section 4.1. For the NAD method equipped with the confidence-based filtering mechanism, we mark it with the † symbol.

### 4.3. Early Stopping Strategy

Early stopping in parallel sampling improves LLM inference by terminating weak reasoning traces and reallocating compute to stronger ones, reducing redundancy. Based on preliminary experiments, we hypothesize that a trace's correctness can be predicted early from its neuron activation patterns. We apply these methods to prune low-quality traces. Specifically, for a partial output $\boldsymbol{y}_{\leq j} = (y_1, y_2, \ldots, y_j)$, we compute the set of neurons activated by $x$ up to the current position $j$ as $N_{\text{activated}}(x, \boldsymbol{y}_{\leq j})$ by Eq.(3), employ the selection schemes from Section 4.1, and resume generation from

| Model | Method | Math Reasoning | | Code Generation | | | Avg. |
|---|---|---|---|---|---|---|---|
| | | AIME24+25 | GPQA | HumanEval | LCBv5 | MBPP | |
| Qwen3-4B-Think | Avg@64 | 74.6 | 66.3 | 96.0 | 61.8 | 84.6 | 76.7 |
| | Cons@64 | 86.7 | 68.2 | – | – | – | – |
| | Short-1@64 | 83.3 | 68.2 | 97.0 | 61.7 | 85.4 | 79.1 |
| | Self-Certainty | 76.7 | 61.1 | – | – | – | – |
| | DeepConf | **88.3** | 70.7 | – | – | – | – |
| | NAD-kNN[†] | 85.0 | **71.7** | 97.0 | **62.3** | **86.4** | **80.5** |
| | NAD-Medoid[†] | 83.3 | 69.7 | **97.6** | 61.7 | 86.0 | 79.6 |
| | NAD-DBSCAN[†] | 85.0 | 71.2 | **97.6** | 61.7 | 86.0 | 80.3 |
| | NAD-MinAct[†] | 83.3 | 69.7 | 95.1 | 58.7 | 85.0 | 78.4 |
| R1-Qwen3-8B | Avg@64 | 71.3 | 60.2 | 92.8 | 57.7 | 83.1 | 73.0 |
| | Cons@64 | 83.3 | **70.2** | – | – | – | – |
| | Short-1@64 | 83.3 | 65.2 | 92.7 | 58.7 | 80.4 | 76.0 |
| | Self-Certainty | 71.7 | 56.1 | – | – | – | – |
| | DeepConf | **85.0** | 67.7 | – | – | – | – |
| | NAD-kNN[†] | **85.0** | 66.7 | 95.1 | 55.1 | 84.4 | 77.2 |
| | NAD-Medoid[†] | **85.0** | 68.7 | 95.7 | **59.3** | 85.4 | **78.8** |
| | NAD-DBSCAN[†] | **85.0** | 67.7 | **96.3** | 58.1 | **85.6** | 78.5 |
| | NAD-MinAct[†] | 81.7 | 60.1 | 95.1 | 56.9 | 83.6 | 75.5 |
| Qwen3-4B-Instruct | Avg@64 | 51.7 | 59.2 | 90.8 | 34.0 | **75.4** | 62.2 |
| | Cons@64 | **65.0** | 61.1 | – | – | – | – |
| | Short-1@64 | 61.7 | 60.6 | 90.9 | 30.5 | 74.3 | 63.6 |
| | Self-Certainty | 51.7 | 61.1 | – | – | – | – |
| | DeepConf | 63.3 | 63.7 | – | – | – | – |
| | NAD-kNN[†] | 58.3 | 66.2 | **92.7** | **35.3** | 75.0 | **65.5** |
| | NAD-Medoid[†] | 53.3 | 65.7 | **92.7** | 34.7 | 75.0 | 64.3 |
| | NAD-DBSCAN[†] | 53.3 | **66.7** | **92.7** | **35.3** | 75.3 | 64.7 |
| | NAD-MinAct[†] | 60.0 | 61.6 | 92.1 | 31.7 | 75.3 | 64.2 |

*Table 1.* Main results of our experiments. Our methods achieve performance competitive with majority voting and consistently surpass sampling average.

the selected trace by itself. We set $j$ equal to the chunk size, $B = 32$ for the experiments. To analyze this phenomenon in a relatively independent manner, we do not apply the confidence-based filtering method in experiments related to early stopping. The choice of the early stopping position will be further discussed in Section 5.3.

## 5. Experiments

### 5.1. Setup

**Models.** We evaluate NAD on three models: Qwen3-4B-thinking-2507, Qwen3-4B-Instruct-2507 (Team, 2025) and DeepSeek-R1-0528-Qwen3-8B (DeepSeek-AI, 2025). For each input, we generate $n = 64$ samples with a temperature of $0.6$ and a top-$p$ value of $0.9$.

**Datasets.** We evaluate NAD on two settings: (1) scientific reasoning with canonical answers, including AIME24, AIME25 (Art of Problem Solving, 2024a;b; 2025a;b) and GPQA (Rein et al., 2024)); and (2) open-ended code generation, including LiveCodeBench v5 (Jain et al., 2024), HumanEval (Chen et al., 2021) and MBPP (Austin et al., 2021), where majority voting is inapplicable.

**Protocol & Baselines.** Under this protocol we report following baselines: (i) **Avg@64** (mean accuracy over all $n$ samples); (ii) **Cons@64** majority vote for tasks with canonical answers (ties count as failure); (iii) **Short-1@64** (Hassid et al., 2025), i.e. select the shortest response; (iv) **Self-Certainty** (Kang et al., 2025); (v) **DeepConf** (Fu et al., 2025). We evaluate under a fixed sampling budget $n = 64$ and a low-interaction regime that mirrors deployments where repeated environment calls are expensive; for code, we adopt a single-execution protocol (only the finally selected candidate is executed once). For More detailed settings, please refer to Appendix C.

### 5.2. Results

The main results are summarized in Table 1, which indicate that: 1) Our approach substantially outperforms baselines in terms of overall performance; 2) On math reasoning datasets with extractable ground-truth answers, our methods demonstrate clear advantages over sampling average while remaining competitive with majority voting based methods, i.e. Cons@64, Self-Certainty and DeepConf; On code generation benchmarks, where existing sample-evaluate-ensemble methods are not applicable, our methods still yield perfor-

| Model | Method | AIME24+25 | | GPQA | | Avg. Acc. |
|---|---|---|---|---|---|---|
| | | Acc. | Token (Δ%) | Acc. | Token (Δ%) | |
| Qwen3-4B-Think | Avg@64 | 74.6 | 55.2 | 66.3 | 102.2 | 70.4 |
| | NAD-kNN | 80.0 | 1.3 (-97.6%) | 67.2 | 2.0 (-98.0%) | 73.6 |
| | NAD-Medoid | **81.7** | 1.3 (-97.6%) | **68.2** | 2.0 (-98.0%) | **75.0** |
| | NAD-DBSCAN | **81.7** | 1.3 (-97.6%) | 65.7 | 2.0 (-98.0%) | 73.7 |
| | NAD-MinAct | 80.0 | 1.2 (-97.8%) | 67.7 | 1.8 (-98.2%) | 73.9 |
| R1-Qwen3-8B | Avg@64 | 70.6 | 48.1 | **58.1** | 99.6 | 64.4 |
| | NAD-kNN | **78.4** | 1.0 (-97.9%) | 56.1 | 1.8 (-98.2%) | 67.3 |
| | NAD-Medoid | 75.0 | 1.1 (-97.7%) | 54.0 | 1.9 (-98.1%) | 64.5 |
| | NAD-DBSCAN | 75.0 | 1.1 (-97.7%) | 54.5 | 1.9 (-98.1%) | 64.8 |
| | NAD-MinAct | 76.7 | 0.9 (-98.1%) | **58.1** | 1.7 (-98.3%) | **67.4** |
| Qwen3-4B-Instruct | Avg@64 | 51.7 | 32.0 | 59.2 | 41.6 | 55.5 |
| | NAD-kNN | 55.0 | 0.4 (-98.8%) | 58.6 | 0.9 (-97.8%) | 56.8 |
| | NAD-Medoid | 55.0 | 0.6 (-98.1%) | 56.6 | 1.0 (-97.6%) | 55.8 |
| | NAD-DBSCAN | 55.0 | 0.6 (-98.1%) | 56.6 | 1.0 (-97.6%) | 55.8 |
| | NAD-MinAct | **61.7** | 0.4 (-98.8%) | **63.1** | 0.9 (-97.8%) | **62.4** |

*Table 2.* Accuracy and token consumption of different methods on scientific reasoning benchmarks after applying early stopping introduced in Section 4.3. Token consumption is reported in millions (M). Our method achieves a two-order-of-magnitude reduction in token usage while maintaining accuracy advantages over random sampling.

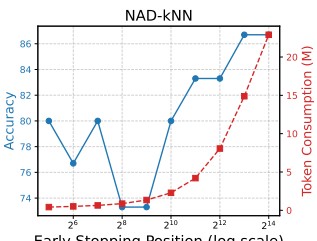

*Figure 4.* Accuracy and token consumption as a function of early stopping position.

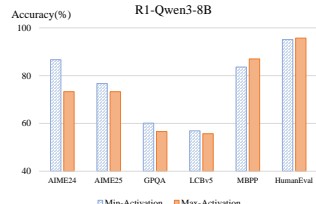

*Figure 5.* Comparison between minimizing and maximizing activated neurons.

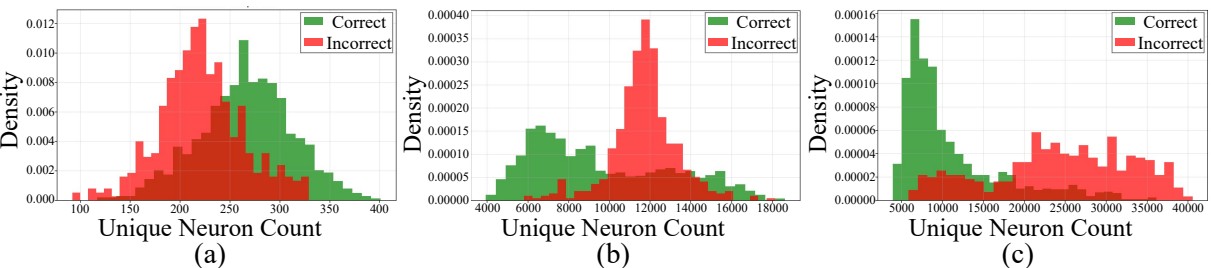

*Figure 6.* Effect of different top-$k$ settings on the distribution of activated neurons for correct and incorrect responses. From left to right: top-$k$ = 2K, 200K, and no top-$k$. The no-top-$k$ setting achieves the best separation.

mance gains over our curated baseline on most tasks; 3) Compared to alternative methods that leverage the global structure across $n$ samples, the minimum-activation method relies solely on the number of activated neurons, which limits its effectiveness; nevertheless, it still significantly surpasses average sampling.

Table 2 reports performance changes and token savings on scientific benchmarks under the early-stopping scheme in Section 4.3; code results appear in Appendix Table 3. Relative to parallel sampling, our method permits stopping after the first chunk, sharply reducing tokens while consistently surpassing random sampling in accuracy. These gains show that early internal neuron activations provide reliable signals of answer quality.

### 5.3. Analysis

**Different implementations for activated neuron computation.** In Section 3.1, we aggregate activated neurons by simply taking the union of all chunks, treating them equally.

To investigate the effect of this aggregate method, we adopt top-$k$ operation (Cao et al., 2025; Wang et al., 2025) to merge neurons across sequences. Specifically, we retain only the neurons with the top-$k$ contribution scores, where $k$ ranges from 2K to 200K, and eventually no top-$k$ filtering is applied (our method). The corresponding distributions of activated neurons are shown in Figure 6. We observe that as $k$ increases: (1) the distribution of activated neurons for correct samples gradually shifts to the left, while that for incorrect samples shifts to the right; (2) the distribution of incorrect samples becomes increasingly uniform. Overall, the distinction between the two distributions becomes more pronounced. We hypothesize that when $k$ is small, the activated neurons focus on high-contribution reasoning paths, losing some finer details. As $k$ increases, more detailed information is incorporated, providing a more comprehensive and discriminative view of the model's internal states.

**The Effect of Early Stopping Position.** In the main experiments, we fix the early stopping position at $B = 32$. Intuitively, the later the truncation point in generation, the

richer the information conveyed by activated neurons. In this section, we investigate how different early stopping positions affect model performance and token consumption. Results across positions ranging from 32 to 16384 are shown in Figure 4. Interestingly, we find that later stopping does not necessarily yield better answer quality (For more results, please refer to Figure 7-Figure 8 in the Appendix). For example, on the AIME24+25 dataset, the kNN method achieves higher accuracy when stopping at the 4096th token compared to using the full response (86.7 vs. 85.0). This phenomenon may be attributed to noise accumulation and signal dilution: as generation proceeds, additional activations may introduce redundancy or errors that obscure the earlier, more reliable signals, leading to degraded answer selection despite longer reasoning traces.

**Relationship between neuron activation and performance.** In Section 3.2, we observed on `AIME24` that correct responses activate fewer neurons than incorrect ones. To further examine this phenomenon, we compare it against the opposite strategy: selecting the reasoning trajectory that maximizes neuron activations. The results of `R1-Qwen3-8B` are reported in Figure 5. On math and science reasoning benchmarks, our hypothesis is confirmed: responses selected based on minimal activations achieve significantly higher accuracy than those based on maximal activations; whereas on code generation tasks, the difference is much less pronounced, and in some cases, the maximal-activation strategy even outperforms the minimal-activation ones. We suggest that this is primarily because code generation is more open-ended, with multiple valid ways to implement the same functionality. To directly test this hypothesis, we conduct a case study on the parenthesis-matching problem, where the model can solve the task either by dynamic programming or by a direct combinatorial formula. Both solutions are correct, yet their activation patterns exhibit low overlap, with a Jaccard similarity of only 0.22. This provides concrete evidence that in open-ended code tasks, multiple valid reasoning paths need not converge to a single activation pattern, which makes the gap between minimal-activation and maximal-activation heuristics less pronounced. Additionally, coding tasks inherently require the model to draw upon a broader range of knowledge, including programming languages, libraries, and domain-specific conventions, which may further weaken the activation–performance link. We hope that further exploration of this line of research can facilitate extending parallel reasoning to more general domains.

**Comparison with Test-Time Training (TTT) method.** We also compare the performance of NAD with Test-Time Training (TTT) methods, since they both do not require explicit, verifiable answers. We train `Qwen3-4B-Instruct` on the AIME24 dataset for 100 steps using the default configuration of TTRL (Zuo et al.,

2025), and the results are shown in Table 6 in the Appendix. The TTRL method yields marginal improvements on the in-domain dataset (AIME24) but exhibits a substantial drop under the OOD setting. Moreover, compared with the inference-time ensemble approach of NAD, TTRL incurs a much higher training cost.

**Effect of data difficulty.** We further examine how data difficulty influences our method. To this end, we conduct experiments on two additional datasets: HMMT25 (HMM, 2025) (a more challenging dataset) and MATH500-L1 (Lightman et al., 2023) (an easier dataset). The results are presented in Table 5. The results show that our method is effective on both easier and more difficult datasets. The standard deviation of accuracy across sampling runs in Table 8 further confirms this: our method consistently improves performance regardless of whether the accuracies are concentrated or dispersed (when the dataset is easy / challenging).

**Computational overhead.** To evaluate the computational overhead of our NAD method, we measured its performance across two stages. During rollout collection, our model inference achieved a high throughput of approximately 2,600–2,700 tokens/s on various benchmarks. Although this generated activation caches as large as 9.4 GB, we utilized memory-mapped files to significantly reduce memory footprint and I/O overhead. In the analysis stage, the complete procedure for 1,920 problems—including Jaccard distance computation and selector algorithm execution—was finished in just 12.74 seconds by leveraging full parallelization, corresponding to a processing throughput of 150.6 problems/second. Further details are provided in Appendix D. In summary, our method does not introduce excessive computational overhead.

## 6. Conclusion

In this work, we analyze LLM internals via neuron-activation patterns in correct vs. incorrect outputs. Correct outputs activate fewer neurons and align more, a reliable quality signal. We propose Neuron-Agreement Decoding (NAD), selecting responses from internal activations. On math, science, and coding, NAD matches majority voting on well-defined tasks and beats average sampling on open-ended coding. Early pruning cuts tokens up to 99% without quality loss. These results show that neuron-level signals improve efficiency and reliability, motivating the ensemble decoding based on internal dynamics.

## 7. Limitations

Despite efficient early selection, open issues remain: (1) **Selector impact**: we introduced several selectors but did not determine which is most effective under different sampling patterns. (2) **Storage overhead**: compute cost is small,

yet storing neuron activations requires substantial disk and memory; more space-efficient representations of internal dynamics are a key direction. Our comparisons use a fixed budget $n = 64$ and a single-execution protocol for code; approaches with many more samples or multiple executions target a different resource regime. Engineering-wise, storage can be non-trivial, but bitset or bitmap encodings and parallel Jaccard under early stopping (@32 token) keep added overhead small. **(3) Interaction between sparsity and agreement.** NAD leverages two complementary signals: activation sparsity (correct trajectories activate fewer neurons) and cross-sample agreement (correct trajectories overlap more in neuron space). A natural question is how these two effects interact. Our interpretation is that they are related rather than independent. Successful trajectories appear to be both more selective and more consistent: they tend to recruit a smaller subset of salient neurons that reflects task-relevant computation. Because this subset is functionally tied to the correct reasoning path, correct trajectories from different samples are also more likely to overlap in neuron space. In this view, activation sparsity is an intra-trajectory selectivity signal — each correct chain focuses its computation on a compact set of neurons — while neuron agreement is an inter-trajectory stability signal — different correct chains converge on the same compact set. Our paper focuses on exploiting these regularities for best-of-$N$ selection and early stopping rather than fully explaining their causal mechanism. Understanding why this pattern arises, and whether one effect partly drives the other, is an interesting direction for future work.

## Impact Statement

This paper presents work whose goal is to advance the field of machine learning. There are many potential societal consequences of our work, none of which we feel must be specifically highlighted here.

## Acknowledgements

This work is supported by New Generation Artificial Intelligence—National Science and Technology Major Project No. 2025ZD0124102.

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

## A. LLM Assistance Disclosure

We used large language model (LLM) tools for grammar and wording refinement during manuscript preparation. We also used image-generation tools to create a stylized alpaca illustration to aid reader understanding. All technical content, analyses, and citations were authored, verified, and remain the sole responsibility of the authors.

## B. Implementation for Threshold Function

In this paper, we adopt a top-$k$ threshold function for key neuron selection, which can be calculated as follows:

**1. Calculate the highest activations on the $j$-th token $y_j$ in each layer $l$:**

$$F_{jl} = \texttt{topk}(A(y_j, l), 64), \tag{6}$$

where $A(y_j, l) \in \mathbb{R}^N$ denotes the contribution score matrix on token $y_j$ in layer $l$, with $[A(y_j, l)]_i = f_{\text{neuron}}(i, l, y_j \mid x \oplus y_{<j})$. Here, $\texttt{topk}(A, k)$ returns the $k$ largest values in $A$.

**2. Find the threshold by aggregating activations across the sequence:**

$$\eta(j, k) = \min\{\texttt{topk}([F_{j1}; F_{j2}; \ldots; F_{jl}], k)\}. \tag{7}$$

In all experiments, we set $k = 500$. Note that the threshold is equivalent to taking the top-$k$ across all activation values on token $y_j$ when 64 in Eq. (6) is scaled up to $N$. We choose to use 64 instead of $N$ for computational efficiency considerations. This token-level thresholding is always applied in our main method. Unless otherwise noted (Sec. 5.3), we do not apply any sequence-level global top-k across tokens; when we do, it is clearly marked as an ablation.

## C. Detailed Experiment Settings

- **Self-Certainty**. We follow the hyperparameter settings from the official repository, using a Borda parameter of $p = 0.5$. The other settings follow what we specified in Section 5.1.

- **DeepConf**. We adopt the Tail Conf-10% configuration, which achieves the best overall performance according to the DeepConf paper. The other settings follow what we specified in Section 5.1.

- **NAD**. In the confidence filtering stage, we set $T_{tail} = 1024$. Since our sampling budget is smaller than that of DeepConf, we slightly increase the filtering ratio to $p\% = 30\%$. For the structure discovery module, we set the kNN parameter to $k = 5$.

| Model | Method | HumanEval | | LCBv5 | | MBPP | | Avg. Acc. |
|---|---|---|---|---|---|---|---|---|
| | | Acc. | Token ($\Delta\%$) | Acc. | Token ($\Delta\%$) | Acc. | Token ($\Delta\%$) | |
| Qwen3-4B-Think | Avg@64 | 97.0 | 52.2 | 58.7 | 191.9 | 85.6 | 169.8 | 80.4 |
| | NAD-kNN | 97.6 | 1.1(-97.9%) | 63.5 | 3.2(-98.3%) | 85.2 | 3.5(-98.0%) | 82.1 |
| | NAD-Medoid | 97.6 | 1.1(-97.9%) | 59.9 | 3.3(-98.3%) | 85.0 | 3.5(-98.0%) | 80.8 |
| | NAD-DBSCAN | 97.6 | 1.1(-97.9%) | 61.1 | 3.3(-98.3%) | 85.2 | 3.6 (-97.9%) | 81.3 |
| | NAD-MinAct | 96.3 | 1.0(-98.1%) | 65.9 | 3.2(-98.3%) | 85.2 | 3.1(-98.2%) | 82.4 |
| R1-Qwen3-8B | Avg@64 | 92.1 | 49.3 | 61.1 | 190.4 | 84.6 | 171.6 | 79.3 |
| | NAD-kNN | 94.5 | 0.9(-98.2%) | 56.9 | 3.2(-98.3%) | 83.2 | 3.3(-98.1%) | 78.2 |
| | NAD-Medoid | 91.5 | 1.0(-98.0%) | 58.7 | 3.2(-98.3%) | 84.0 | 3.5(-98.0%) | 78.1 |
| | NAD-DBSCAN | 91.5 | 1.0(-98.0%) | 61.7 | 3.2(-98.3%) | 83.0 | 3.5(-98.0%) | 78.7 |
| | NAD-MinAct | 93.9 | 0.9(-98.2%) | 58.7 | 2.9(-98.4%) | 82.4 | 3.0(-98.2%) | 78.3 |
| Qwen3-4B-Instruct | Avg@64 | 89.6 | 6.0 | 31.1 | 26.6 | 74.9 | 37.1 | 65.2 |
| | NAD-kNN | 90.2 | 0.4(-93.3%) | 35.9 | 0.6(-97.7%) | 73.9 | 1.4(-96.2%) | 66.7 |
| | NAD-Medoid | 90.9 | 0.4(-93.3%) | 36.5 | 0.7(-97.4%) | 74.7 | 1.5(-96.0%) | 67.4 |
| | NAD-DBSCAN | 90.9 | 0.4(-93.3%) | 35.9 | 0.7(-97.4%) | 76.2 | 1.5(-96.0%) | 67.7 |
| | NAD-MinAct | 92.1 | 0.4(-93.3%) | 31.7 | 0.6(-97.7%) | 77.0 | 1.4(-96.2%) | 66.9 |

*Table 3.* Accuracy and total token consumption of different methods on code benchmarks after applying early stopping introduced in Section 4.3. Token consumption is reported in millions (M). Our method achieves a two-order-of-magnitude reduction in token usage while maintaining accuracy advantages over random sampling.

# D. Computational Cost

We evaluated the computational overhead of NAD using two NVIDIA H20 GPUs with a tensor parallelism size of 2, running the R1-Qwen3-8B model at a batch size of 64. For AIME24 and AIME25 (each comprising 30 problems with 64 samples per problem), the model generated 44,223,651 and 41,083,660 tokens respectively, achieving inference throughputs of approximately 2,711.6 and 2,730.7 tokens per second. The corresponding activation caches were 2.0 GB and 1.8 GB, respectively. For the larger LiveCodeBench-v5 benchmark (167 problems × 64 samples), the model produced 190,542,489 tokens at a throughput of 2,603.8 tokens per second, generating an activation cache of 9.4 GB. All neuron activations were stored on disk using memory-mapped files, significantly reducing resident memory usage and I/O overhead, while facilitating efficient random-access retrieval during subsequent analysis.

In the NAD analysis stage, we conducted experiments on a total of 1,920 problems, each consisting of 64 samples. For each problem, we computed the full pairwise Jaccard distances between neuron activation sets of the 64 samples, then executed the NAD selector algorithm on the resulting distance matrices. This entire analysis procedure leveraged full parallelization across problems and was completed within 12.74 seconds, corresponding to a processing throughput of 150.6 problems per second. Each analysis process typically occupied around 2.9 GB of memory.

| Model | Method | Math Reasoning | | Code Generation | | | Avg. |
|---|---|---|---|---|---|---|---|
| | | AIME24+25 | GPQA | HumanEval | LCBv5 | MBPP | |
| | Avg@64 | 14.2 | 51.5 | 87.6 | 26.5 | 76.2 | 51.2 |
| | Cons@64 | **18.3** | 55.0 | – | – | – | – |
| Qwen2.5-72B-Instruct | NAD-kNN[†] | **18.3** | 54.5 | 88.4 | 25.8 | 76.8 | 52.8 |
| | NAD-Medoid[†] | **18.3** | 54.0 | 87.8 | **28.1** | **77.2** | 53.1 |
| | NAD-DBSCAN[†] | **18.3** | 54.0 | **89.0** | 27.5 | 77.0 | **53.2** |
| | NAD-MinAct[†] | 11.7 | **57.6** | 87.8 | 26.4 | 76.3 | 51.9 |

*Table 4.* Results of `Qwen2.5-72B-Instruct` on all datasets. The results demonstrate that our method remains effective on larger models.

| Model | Method | HMMT25 | MATH500-L1 |
|---|---|---|---|
| | Avg@64 | 50.5 | 99.1 |
| | Cons@64 | 60.0 | **100.0** |
| Qwen3-4B-Think | NAD-kNN[†] | 56.7 | **100.0** |
| | NAD-Medoid[†] | 60.0 | **100.0** |
| | NAD-DBSCAN[†] | 60.0 | **100.0** |
| | NAD-MinAct[†] | **66.7** | **100.0** |
| | Avg@64 | 48.7 | 99.2 |
| | Cons@64 | **63.3** | **100.0** |
| R1-Qwen3-8B | NAD-kNN[†] | **63.3** | **100.0** |
| | NAD-Medoid[†] | 60.0 | 97.7 |
| | NAD-DBSCAN[†] | 60.0 | 97.7 |
| | NAD-MinAct[†] | 56.7 | 97.7 |
| | Avg@64 | 29.7 | **97.7** |
| | Cons@64 | 30.0 | **97.7** |
| Qwen3-4B-Instruct | NAD-kNN[†] | 30.0 | **97.7** |
| | NAD-Medoid[†] | **36.7** | **97.7** |
| | NAD-DBSCAN[†] | 33.3 | **97.7** |
| | NAD-MinAct[†] | 30.0 | 95.4 |

*Table 5.* Ablation studies on math datasets of varying difficulty (HMMT25 (HMM, 2025) and MATH500-L1 (Lightman et al., 2023)). The results show that our method is effective on both easy and hard datasets.

# E. Comparison with Test-Time Training (TTT) Method

In this section, we compare the performance of NAD with Test-Time Training (TTT) methods. We train `Qwen3-4B-Instruct` on the AIME24 dataset for 100 steps using the default configuration of TTRL (Zuo et al., 2025), and the results are shown in Table 6. The TTRL method yields marginal improvements on the in-domain dataset (AIME24) but exhibits a substantial drop under the OOD setting. Moreover, compared with the inference-time ensemble

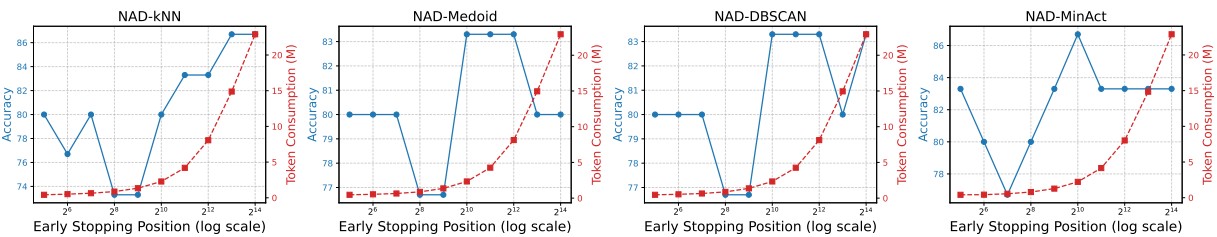

*Figure 7.* Accuracy and token consumption as a function of early stopping position of `R1-Qwen3-8B` on `AIME24`.

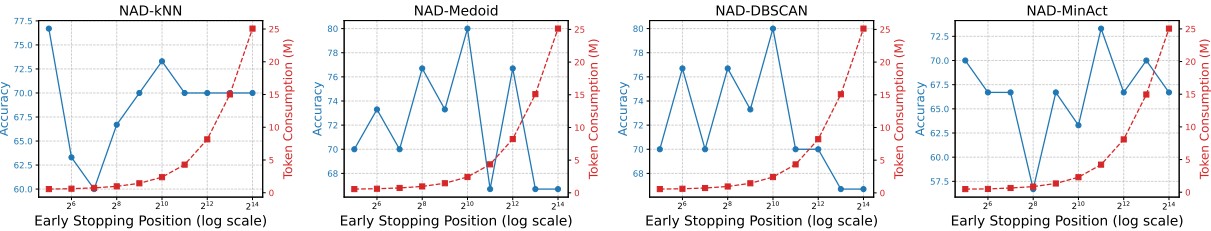

*Figure 8.* Accuracy and token consumption as a function of early stopping position of `R1-Qwen3-8B` on `AIME25`.

approach of NAD, TTRL incurs a much higher training cost.

## F. Effect of Data Difficulty

In this section, we further examine how data difficulty influences our method. To this end, we conduct experiments on two additional datasets: HMMT25 (HMM, 2025) (a more challenging dataset) and MATH500-L1 (Lightman et al., 2023) (an easier dataset). The experimental settings follow those in the main paper, and the results are presented in Table 5. The results show that our method is effective on both easier and more difficult datasets. The standard deviation of accuracy across sampling runs in Table 8 further confirms this: our method consistently improves performance regardless of whether the accuracies are concentrated or dispersed (when the dataset is easy / challenging).

## G. Hyperparameters and Implementation Details

This section summarizes the main hyperparameters and implementation choices used in our experiments, and links them to the corresponding ablation tables and figures.

**Sampling and decoding.** Unless otherwise stated, we draw $n = 64$ trajectories per problem for all models with temperature $\tau = 0.6$ and top-$p = 0.9$ (see Section 5.1). Appendix Table 7 sweeps $\tau \in \{0.3, 0.6, 0.9\}$ for R1-Qwen3-8B on the AIME dataset and shows that all NAD variants consistently outperform AVG@64 and remain competitive with CONS@64 across

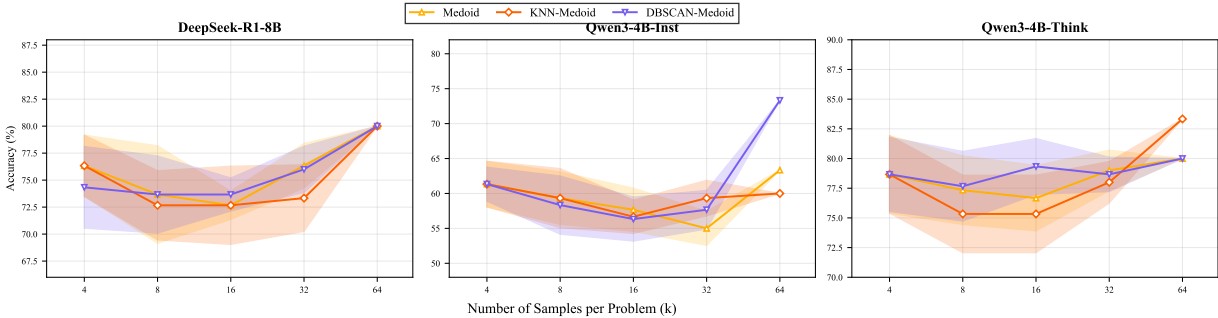

*Figure 9.* Results on AIME24 under different sampling counts $n$ estimated via bootstrap, where the shaded regions denote the bounds of the 95% confidence interval (CI). As $n$ increases, the ensemble accuracy shows an upward trend and generally surpasses sampling average for larger $n$, demonstrating the robustness of our method.

| Method | AIME24 | AIME25 | GPQA | HumanEval | LCB-v5 | MBPP |
|---|---|---|---|---|---|---|
| Avg@64 | 58.8 | 44.7 | 59.2 | 90.8 | 34.0 | **75.4** |
| TTRL | 58.9 | 37.8 | 59.8 | **92.9** | 33.3 | 74.3 |
| NAD-kNN† | **66.7** | 50.0 | 66.2 | 92.7 | **35.3** | 75.0 |
| NAD-Medoid† | 63.3 | 43.3 | 65.7 | 92.7 | 34.7 | 75.0 |
| NAD-DBSCAN† | 63.3 | 43.3 | **66.7** | 92.7 | **35.3** | 75.3 |
| NAD-MinAct† | **66.7** | **53.3** | 61.6 | 92.1 | 31.7 | 75.3 |

*Table 6.* Performance comparison between NAD and TTRL. TTRL offers only marginal in-domain gains, suffers significant OOD degradation, and incurs substantially higher training cost than NAD.

| Method | $\tau = 0.3$ | $\tau = 0.6$ | $\tau = 0.9$ |
|---|---|---|---|
| Avg@64 | 70.7 | 71.3 | 69.3 |
| Cons@64 | 81.7 | 83.3 | 80.0 |
| NAD-kNN | 83.3 | 85.0 | 80.0 |
| NAD-Medoid | 81.7 | 85.0 | 76.7 |
| NAD-DBSCAN | 81.7 | 85.0 | 78.3 |
| NAD-MinAct | 78.3 | 81.5 | 80.0 |

*Table 7.* Ablation of temperature of R1-Qwen3-8B on AIME dataset. Our method achieves consistent improvements across different temperature ($\tau$) settings.

| Model | Math Reasoning | | | Code Generation | | |
|---|---|---|---|---|---|---|
| | AIME24 | AIME25 | GPQA | HumanEval | LCBv5 | MBPP |
| Qwen3-4B-Think | 4.6 | 4.6 | 1.9 | 1.1 | 2.3 | 0.5 |
| R1-Qwen3-8B | 4.5 | 5.4 | 2.3 | 1.4 | 2.0 | 0.9 |
| Qwen3-4B-Instruct | 5.2 | 5.5 | 2.2 | 1.4 | 1.7 | 7.8 |

*Table 8.* Standard deviation of accuracy across sampling runs. The results demonstrate that NAD is effective when accuracy is concentrated (e.g., HumanEval) as well as when it is dispersed (e.g., AIME).

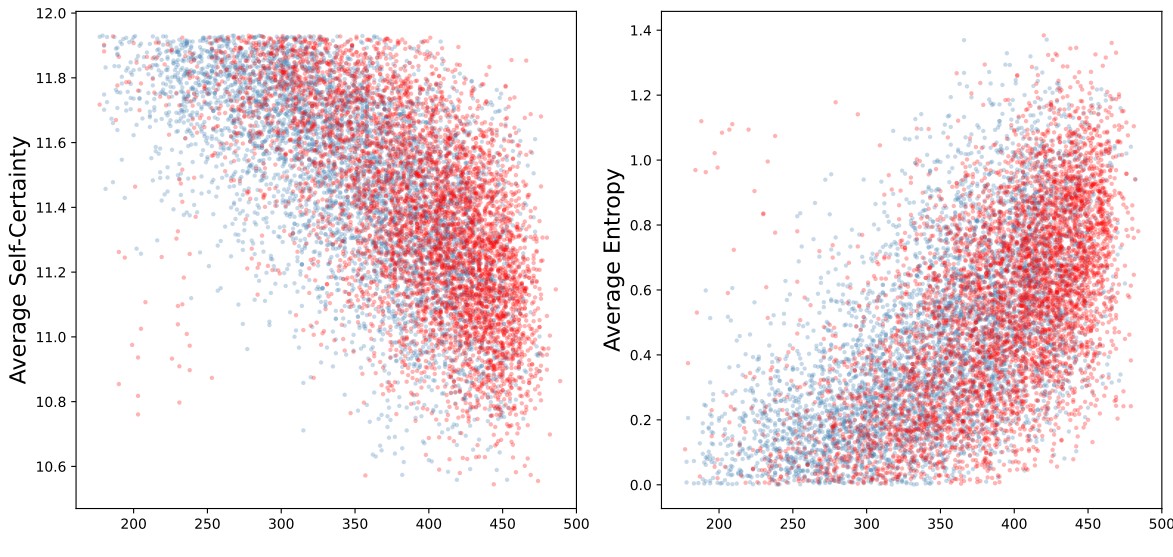

*Figure 10.* A more detailed version of Figure 1a. The blue points represent correct samples, while the red points represent incorrect samples. The results show no significant difference in their overall trends, and the discrepancy in neuron counts further supports our Insight 2.

temperatures. This indicates that neuron-agreement decoding is robust to reasonable changes of the sampling temperature.

**Early stopping.** For early-stopping experiments (Section 4.3), we fix the chunk size to $B = 32$ tokens and apply NAD after observing the first chunk. Only the selected trajectory is allowed to continue decoding, while the remaining ones are terminated. Table 3 in the main text and Appendix Table 3 report the resulting accuracy and total token consumption on scientific reasoning and code benchmarks, respectively, showing roughly two orders of magnitude reduction in tokens with minimal or no loss in accuracy. Appendix Figures 7 and 8 further plot accuracy and token usage as a function of the stopping position for R1-Qwen3-8B on AIME24 and AIME25, demonstrating that stopping after 32–64 tokens already provides a strong trade-off between performance and efficiency, while delaying the stopping point yields diminishing or even slightly negative returns.

**Number of samples.** Appendix Figure 9 varies the number of samples $n \in \{4, 8, 16, 32, 64\}$ on AIME24 for all three backbone models. As $n$ increases, the ensemble accuracy of NAD-based selectors improves and generally surpasses the sampling average for larger $n$, while the bootstrap confidence intervals remain tight. These results support our choice of $n = 64$ as a reasonable budget that balances compute and performance.

