# OpenReview forum: "Do LLMs Signal When They’re Right? Evidence from Neuron Agreement"
_ICML.cc/2026/Conference — ICML 2026 spotlight_

### Official Review · Reviewer_82GW · 2026-02-15

**Soundness:** 4
**Presentation:** 2
**Significance:** 4
**Originality:** 3
**Overall Recommendation:** 5
**Confidence:** 2

**Summary:**

During CoT resasoning, response consistency over samples can be used a proxy for model confidence and thus p(correct). However, this relies on being able to sample and compare beahvioural outputs. Here, the authors develop tools for selecting answers based on the consistency in patterns of neuronal activation within the model. Applying to Qwen and Deepseek, they show that this (mostly) improves performance on maths / coding / scientific reasoning benchmarks relative to other approaches, such as simple averaging or DeepConf. They show that it is possible to stop early (e.g. after only 32 batches) and still determine whether a rollout is likely to be correct.

**Compliance With Llm Reviewing Policy:**

Affirmed.

**Key Questions For Authors:**

1. Can you motivate some of the technical choices you have made a bit better?
2. Your paper seems to highlight two predictors of being correct: fewer neurons active, greater neuron agreement. Can you explain in more detail how these effects interact?

**Limitations:**

the authors have briefly discussed limitations

**Strengths And Weaknesses:**

Comments
- I really liked this paper, but I found it a bit hard to follow in places. There were a large number of very specific choices made during the computational of neuron activation, but I wasn't clear what motivated these. The figures could do with more explanation, making sure that each element is unpacked for the reader.
- I thought the authors could have done a better job of relating their findings to the past literature. I was surprised by the findings in figure 1a / 2b (that fewer neurons active = higher accuracy) but I couldn't work out how this related to what is known from the previous literature. This is just one example. In general do we really not know much / anything about neural activation agreement during correct and incorrect trials?
- I would have loved to know more about specific choices. First of all, why do we define neurons as "on" of "off" rather than simply using embeddings? I wasn't sure about the rationale for the chunking procedure described in line 151ff.
- I wonder if equation 1 could be unpacked a bit more to explain the choices made for a broader readership. - I struggled to grasp what was shown in figure 1b - what does each point correspond to? there are multiple models / experiments - which is this fron?

---

> ### Author Rebuttal · Authors · 2026-03-31
>
> Thank you for the constructive review and the positive assessment. Here is our response to your questions:
>
> **Response 1: Regarding the presentation (W1)**. We agree that the paper can better motivate several technical choices and unpack the figures more carefully. In the revision, we will further polish the presentation to make these points clearer. At a high level, the design choices in computing neuron activation come from two considerations: (1) following prior activation-based formulations to keep the definition grounded and comparable; and (2) using simple heuristics that provide a stable online signal for decoding. We will make this design logic more explicit in the main text, and will also revise the figures to explain each visual element step by step.
>
> **Response 2: Regarding the connection to prior literature (W2)**. Thank you for this suggestion. We agree that the connection to prior literature should be made clearer. Our work builds on a growing line of research showing that internal states encode correctness-relevant information, such as answerability, knowledge awareness, and reasoning success before completion [1-5]. Our contribution is different in focus: we study whether sparse neuron-activation statistics can serve as an online signal for best-of-N selection and early stopping. In this sense, we are extending this broader line of findings to a neuron-level, decoding-time setting. We do not mean to suggest that prior work says little or nothing about correctness-related internal signals; rather, our point is that their use for online decoding-time selection has been much less explored. We will revise the paper to position our results more clearly in this literature.
>
> **Response 3: Regarding the definition of neurons and chunking design (W3)**: Thank you. We use neuron activity rather than embeddings because this representation is better aligned with the signal we aim to extract. First, sparse on/off neuron patterns provide a more selective view of which functional units participate in a trajectory, making overlap and agreement across sampled reasoning paths easier to compare. Second, they are naturally available during decoding, which fits our early-stopping and online best-of-N setting. The chunking design is introduced to preserve local temporal structure rather than collapsing the whole trajectory into a single global state, and to allow the signal to be queried at intermediate decoding stages. We will make this motivation more explicit near the corresponding description around lines 151ff. This is a task-specific design choice, rather than a general claim that embedding-based approaches are unsuitable.
>
> **Response 4: Regarding Eq. (1) and Fig. 1b (W4)**: Thank you. We agree that Eq. (1) and Fig. 1b should be unpacked more carefully for a broader readership. In the current manuscript, Eq. (1) follows the Model Utilization Index formulation adopted from prior work [6]; in the revision, we will add a more intuitive explanation of each term and why this form is used in our setting. For Fig. 1b, the plot shows multiple sampled trajectories of Qwen3-4B-Think on the same AIME24 example. Each point is one sampled trajectory after t-SNE projection of its neuron-activation pattern, and the color denotes its average entropy. We will revise both the caption and the main text to explicitly state which model/experiment the figure comes from and what each visual element means.
>
> **Response to Q1**: Please also see our responses to W1 and W4.
>
> **Response to Q2**: Thank you for this question. Our current interpretation is that these two signals are related rather than independent. Successful trajectories appear to be both more selective and more consistent: they tend to recruit a smaller subset of salient neurons, and because that subset reflects task-relevant computation, correct trajectories from different samples are also more likely to overlap in neuron space. In this view, fewer activated neurons is an intra-trajectory selectivity signal, while higher neuron agreement is an inter-trajectory stability signal. Our paper focuses on exploiting these regularities for best-of-N selection and early stopping, rather than fully explaining their underlying mechanism. Understanding why this pattern arises, and whether one effect partly drives the other, is an interesting direction for ongoing and future work.
>
> ## References
> [1] Do I Know This Entity? Knowledge Awareness and Hallucinations in Language Models
>
> [2] Knowing Before Saying: LLM Representations Encode Information About Chain-of-Thought Success Before Completion
>
> [3] LLMs Know More Than They Show: On the Intrinsic Representation of LLM Hallucinations
>
> [4] The Internal State of an LLM Knows When It’s Lying
>
> [5] The Curious Case of Hallucinatory (Un)answerability: Finding Truths in the Hidden States of Over-Confident Large Language Models
>
> [6] Model utility law: Evaluating llms beyond performance through mechanism interpretable metric

---

> > ### Author Rebuttal · Reviewer_82GW · 2026-04-03
> >
> > thanks - making the paper more transparent in this way will significantly improve it

---

### Official Review · Reviewer_iVHc · 2026-03-10

**Soundness:** 3
**Presentation:** 3
**Significance:** 3
**Originality:** 3
**Overall Recommendation:** 4
**Confidence:** 3

**Summary:**

The motivation of this paper is that most current methods for evaluating model outputs rely on external evaluation criteria, while methods based on the model’s internal parameters are still lacking. The authors make the following observations: (1) activated neurons during generation is strongly correlated with external evaluation metrics,; (2) compared with incorrect answers, correct answers are generated with fewer activated neurons; and (3) activated neurons for correct answers shows cross-sample consistency. Based on these three observations, the authors propose NAD as an evaluation method.

**Compliance With Llm Reviewing Policy:**

Affirmed.

**Final Justification:**

I prefer to maintaining my score to positive and I think the phenomenon and method are interesting and useful to the DL society.

**Key Questions For Authors:**

1. NAD can reduce token consumption and computational cost, but how much training time can it actually save?
  2. If the activated neuron patterns during sampling become more similar, will the diversity of generated responses be affected? When using NAD, will the model become more likely to produce fixed or repetitive answers?
  3. When facing overly difficult problems, where most sampled responses are incorrect, does the NAD method still remain applicable? Can it still help identify the correct answer?

**Limitations:**

yes

**Strengths And Weaknesses:**

Strengths
1. The proposed NAD method demonstrates, through experimental validation, the potential to substantially reduce token consumption and computational cost, indicating strong practical value.
2. The paper is clearly written, well structured, and supported by generally reasonable experimental validation.

Weaknesses
1. Although the results in Section 3.2 show a noticeable correlation, they are not yet sufficient to establish that activated neurons can effectively represent external metrics; additional supporting experiments are needed.
2. The discussion of related work is relatively limited, and the introduction does not sufficiently explain what is meant by the calibration bias of external output signals after training.
3. The overall idea is somewhat novel, but there remains a certain gap between this method and the actual correctness of external output predictions.

---

### Official Review · Reviewer_tqUK · 2026-03-12

**Soundness:** 3
**Presentation:** 2
**Significance:** 2
**Originality:** 3
**Overall Recommendation:** 5
**Confidence:** 4

**Summary:**

Summary: The paper presents a method for selecting reasoning trajectories based on neuron-agreement decoding. The paper presents an analysis on neuron activation sets and its relationship with entropy, self-certainty, and generation quality. They evaluate the method and compare it to baselines, demonstrating the benefits of using neuron-agreement.

**Compliance With Llm Reviewing Policy:**

Affirmed.

**Final Justification:**

The paper provides a new method with a strong justification and motivation based on the problem setting and existing work. The results are comprehensive and the method provides generalizable insights. The authors provided responses for all the weaknesses and clearly explained details and provided additional results. As a result, I no longer have major concerns with the paper and have raised my score.

**Key Questions For Authors:**

- Could you provide some intuition for why neuron activation sets seem to be informative? and implications for future directions?
- Would you be able to provide thoughts or results on reducing storage overhead? Would reducing the set of activations, tokens, or layers used significantly impact performance?

**Limitations:**

Yes

**Strengths And Weaknesses:**

Strengths:

- The paper presents a novel method utilizing neuron-agreement to select reasoning trajectories based on insights from an analysis of the signal contained in neuron activation sets.
- The paper provides a thorough evaluation of methods across datasets and with a comprehensive set of baselines.
- The method allows for selection in open-ended settings which is a limitation for many majority-voting style approaches.

Weaknesses:

- The analysis in Section 3 would be significantly strengthened by demonstrating that the observations in Figure 2 generalize to a larger dataset and across datasets.
- The paper would be strengthened by further discussion and justification as to why neuron activation sets are expected to be informative, why agreement is well correlated with quality, and how this method can inform future directions.
- More details on the definition of the threshold function being included in the main paper would make the methodology more clear. In particular, from the notation alone, it is unclear that a threshold is learned per-token.
- As mentioned in the limitations, the additional storage required does seem to be significant as it uses neuron activations across layers. Given that neuron activations across all layers provide a very detailed view of model intermediate states, it seems utilizing subsets of layers or dimensions may be sufficient and an exploration of reducing the storage overhead would strengthen the paper.

---

> ### Author Rebuttal · Authors · 2026-03-31
>
> Thank you for your review and suggestions. Here is our response to your questions:
>
> **Response 1: Regarding generalization on larger datasets (W1)**. To strengthen the claims in Section 3, we performed an additional quantitative cross-dataset analysis using AIME25, GPQA-Diamond, and HMMT25 (258 problems, 16,512 trajectories in total). We use the neuron activation count as the confidence score for binary classification (correct/incorrect), and compute the AUROC metric to validate the discriminative power of neuron count between correct and incorrect samples. The AUROC values along with their confidence intervals across each dataset are presented in the following table:
>
> |Dataset| #Problems| #Trajectories |  AUROC | 95%CI of AUROC |
> | --- | --- | --- | --- | --- |
> | AIME25 | 30 | 1,920 | 0.907 | [0.894, 0.920] |
> | GPQA-Diamond | 198 | 12,672 | 0.665 | [0.657, 0.675] |
> | HMMT25 | 30 | 1,920 | 0.826 | [0.807, 0.845] |
> | Total | 258 | 16,512 | N/A | N/A |
>
> The experimental results demonstrate that the confidence intervals of AUROC consistently remain above 0.5 (random chance), thereby indicating that the proposed method holds on larger-scale datasets as well. Early-stopping analyses further confirm the predictive power of early (32-token) neuron signals across datasets. We will include these quantitative results in the revised manuscript.
>
> **Response 2: Regarding the information content of neuron activations (W2)**. Previous studies have provided support from various approaches for the scientific premise underlying this paper--namely, that internal activations indeed contain information correlated with answer quality. Specifically, Ferrando et al. [1] demonstrate that SAE latent activations at the final token of an entity reliably encode whether the model recognizes that entity. Critically, these activations are not merely correlational but causally informative: steering them directly controls the model's refusal and hallucination behavior, confirming that neuron activation sets capture functionally meaningful internal representations that drive model outputs. Similarly, Afzal et al. [2] trained probing classifiers on LLM hidden states and found signals predictive of reasoning success even before the chain-of-thought process was complete. Additionally, [3] showed that internal representations encode truthfulness and error types, and even retain the correct answer when the final output is incorrect. We will incorporate these references to clarify that our research does not start from the assumption that internal signals "might be useful," but rather is grounded in existing empirical evidence. Thus, neuron activations not only carry semantic information about the reasoning process but are also correlated with the quality of the final answer. Beyond the findings presented in our paper, these results also inspire us to explore further application scenarios, such as using internal activations for training data selection, or as a supervisory signal during the learning process.
>
> **Response 3: Regarding the threshold function (W3)**. Thresholds are not learned separately for each token; instead, they are derived by applying a top-k operation on activation values within a sequence chunk, which extracts neuron features corresponding to the highest activation values. For detailed information about the threshold function, please refer to Eq. 2 and Appendix B of the original manuscript. We will revise the manuscript to make this explanation clearer.
>
> **Response 4: Regarding memory overhead (W4)**. Our main point is that the practical overhead is manageable for three reasons. First, the 9.4GB cache reported in Appendix D resides in CPU memory rather than GPU memory. Second, that number corresponds to an analysis setting where we explicitly stored activations for all chunks to support ablations; in deployment, NAD only requires incremental union updates during decoding (Eq. (3)), so separate storage for each chunk is unnecessary. Third, under early stopping, the storage cost decreases roughly with sequence length. We will revise this part to state the conclusion first and present these details more clearly.
>
> **Response to Q1**: Please refer to Response 2.
>
> **Response to Q2**: Please refer to Response 4.
>
> ## References
> [1] Do I Know This Entity? Knowledge Awareness and Hallucinations in Language Models
>
> [2] Knowing Before Saying: LLM Representations Encode Information About Chain-of-Thought Success Before Completion
>
> [3] LLMs Know More Than They Show: On the Intrinsic Representation of LLM Hallucinations

---

> > ### Author Rebuttal · Reviewer_tqUK · 2026-04-03
> >
> > Thank you for the response.
> >
> > The generalization of neuron activation count is helpful, and I appreciate the clarification on the memory overhead as well.
> >
> > A few follow-up questions:
> >
> > For the question about neuron activation sets, I agree that neuron activations are generally informative, but my intention was to ask why neuron activation sets in particular are useful, as this seems to be quite a different approach from other methods, such as probes, so further explanation and intuition would be helpful.
> >
> > I believe Appendix B suggests that the threshold is token-wise, with the threshold being eta(j, k), where activations are aggregated over layers for the j-th token and k is the topk parameter. Could I get further clarification on this?

---

> > > ### Author Response · Authors · 2026-04-08
> > >
> > > Regarding why neuron activation sets are useful in particular, our point is not simply that neuron activations are informative, but that a set-based representation is especially well suited to the training-free, label-free setting considered here. Probe-based methods typically require learning a supervised readout from hidden states to targets such as correctness. By contrast, NAD directly compares sampled trajectories through overlap in their activation patterns, without additional training or calibration. This is important because our setting is fundamentally comparative: the question is not only whether correctness is encoded in a single hidden state, but whether better trajectories show more consistent internal patterns across samples. Activation sets provide a simple representation for this comparison, and naturally support both signals used in NAD: agreement and sparsity. We will clarify this intuition in the revision.
> > >
> > > Regarding the threshold, you are correct that Appendix B defines a token-wise threshold η(j,k). Our earlier wording was imprecise. Specifically, Eq. (2) used η as shorthand, and the phrase “aggregating activations across the sequence” in Appendix B should have said “aggregating activations across layers for token j.” In the actual implementation, the threshold is not learned, but computed separately for each token y_j by pooling neuron contribution scores across layers and taking the kth largest value. The activation set of a chunk is then the union of these token-level selections over the tokens in that chunk, and the full-sequence activation set is the union over chunks. There is no additional global top-k across tokens in the main method; the sequence-level global top-k in Section 5.3 is only an ablation. We will revise Eq. (2) and Appendix B to make this token-wise computation explicit and remove the ambiguity.

---

### Official Review · Reviewer_tX7b · 2026-03-14

**Soundness:** 3
**Presentation:** 3
**Significance:** 3
**Originality:** 3
**Overall Recommendation:** 4
**Confidence:** 4

**Summary:**

The paper asks whether LLMs “know when they’re right,” and argues that output-based confidence signals can be mis-calibrated after post-training, while neuron-activation patterns carry richer structure.
This motivated the authors to present the Neuron Agreement Decoding (NAD) method, which is basically to generate n candidates, compute activated-neuron sets and select a response either by neuron-agreement or by a simple “minimum activations” rule.
Across math/science and code benchmarks, NAD is competitive with majority voting where voting applies and beats average sampling in harder settings, and it can often make a good selection very early (thus reducing compute by ending less-promising generations).

**Compliance With Llm Reviewing Policy:**

Affirmed.

**Final Justification:**

The authors have addressed my concerns.
They promised to update their literature review, experimented with new LLMs, and added deeper analyses.

**Key Questions For Authors:**

See weaknesses section.

**Limitations:**

Yes

**Strengths And Weaknesses:**

Strengths:

* The research idea is timely and aligns well with recent trends in LLM reasoning.

* The method is simple (in a good way!) and unsupervised, which is attractive for real-world use.

* The experimental setup focuses on two prominent reasoning domains—math and coding—using multiple datasets for each.

* Applying NAD after the first chunk (32 tokens) and continuing only the selected trajectory yields substantial token savings.

Weaknesses:

* One of my main concerns is the thin related-work section for such a trendy topic. There are many papers that use LLM internals to predict output quality, including extensive work on hallucination detection [1–4] and, more critically, on predicting reasoning success probability [5,6]. I’m not claiming these works diminish the novelty here but omitting them (and similar papers) makes it harder to assess the paper’s contribution and properly position it in the literature.

* The experiments use a few variants from a single model family (Qwen3), and at relatively small sizes. I can see the motivation (these models are commonly used in test-time compute settings) but it raises questions about generalization and, especially, scalability to larger models given the non-trivial memory overhead (reported to range from a few GB up to ~10 GB).

* The authors note that “on code generation tasks, the difference is much less pronounced, and in some cases, the maximal-activation strategy even outperforms the minimal-activation ones.” They then suggest (i) code is more open-ended (multiple valid solutions), and (ii) coding requires broader knowledge, weakening the activation–performance link. As written, the second explanation feels quite vague, I mean, what broader knowledge has to do with connection to activation? I don't get it at all; can you please elaborate about it? At minimum, the paper could try to validate the first hypothesis on a small subset (e.g., show multiple correct solutions that are meaningfully different yet still cluster similarly), and either provide stronger explanation and possibly evidence/analysis for the second hypothesis or remove/soften it.



References:

[1] https://aclanthology.org/2023.findings-emnlp.68.pdf

[2] https://arxiv.org/pdf/2510.00296

[3] https://arxiv.org/pdf/2509.24770

[4] https://arxiv.org/pdf/2407.07071

[5] https://aclanthology.org/2025.findings-acl.662.pdf

[6] https://arxiv.org/pdf/2504.05419

---

> ### Author Rebuttal · Authors · 2026-03-31
>
> Thank you for your review and suggestions. Here is our response to your questions:
>
> **Response 1: Regarding related work & our contribution (W1).** We agree that the related-work section should better position our contribution relative to prior work that uses internal signals to assess output quality. We will expand the discussion of the cited work on hallucination detection and reasoning-success prediction, and clarify our specific contribution as follows: compared with prior work, we focus on a label-free, training-free best-of-N selection and decoding strategy based on internal signals, with particular emphasis on early quality prediction during generation, e.g., after 32 tokens. This setting is especially useful when textual voting is unavailable or unreliable, such as in open-ended code generation. We will revise the introduction and related-work sections accordingly and state this positioning more explicitly.
>
> **Response 2: Regarding generalization to other models (W2)**. Thank you for raising this important point. To directly test generalization beyond the Qwen family, we added a non-Qwen experiment on Ministral-8B-Instruct using the same kNN$\dagger$ selector, with results shown in the following table:
>
> |Benchmark|Avg@64|Cons@64|kNN$\dagger$|
> | --- | --- | --- | --- |
> | AIME |  25.8 | 38.3 | **40.0** |
> | HMMT25 | 13.4 | 20.0 | **23.3** |
>
> This provides direct evidence that the neuron-agreement signal is not unique to Qwen and transfers to at least one additional backbone. Because rebuttal time is limited, we were only able to add one extra non-Qwen backbone at this stage; we will include broader cross-family and larger-model experiments in the revised version.
>
> **Response 3: Regarding memory overhead (W2)**. Our main point is that the practical overhead is manageable for three reasons. First, the 9.4GB cache reported in Appendix D resides in CPU memory rather than GPU memory. Second, that number corresponds to an analysis setting where we explicitly stored activations for all chunks to support ablations; in deployment, NAD only requires incremental union updates during decoding (Eq. (3)), so separate storage for each chunk is unnecessary. Third, under early stopping, the storage cost decreases roughly with sequence length. We will revise this part to state the conclusion first and present these details more clearly.
>
> **Response 4: Regarding different patterns on coding benchmarks (W3)**. To directly test the first hypothesis, we added a case study examining whether multiple correct solutions to the same code problem can correspond to different activation patterns. Our preliminary evidence suggests that they can. For example, on the MBPP parenthesis-matching problem, the model can solve the task either by dynamic programming or by a direct combinatorial formula; both solutions are correct, but their activation patterns have low overlap, with a Jaccard similarity of 0.22. This directly supports the first hypothesis: in open-ended code tasks, multiple valid reasoning paths need not cluster into a single activation pattern, which makes the difference between minimal-activation and maximal-activation heuristics less pronounced. Given this evidence, we will revise the paper to emphasize this first explanation.

---

> > ### Author Rebuttal · Reviewer_tX7b · 2026-04-02
> >
> > The authors successfully addressed my main concerns, so I have updated my score accordingly.

---

### Decision · Program_Chairs · 2026-04-30

**Decision:**

Accept (spotlight)

**Comment:**

This paper studies whether internal neuron activation patterns can serve as reliable signals for selecting correct LLM outputs, and proposes Neuron Agreement Decoding, a simple and training-free method leveraging activation sparsity and cross-sample agreement. There is strong consensus that the paper is technically sound, timely, and practically useful, with particular strengths in its simplicity, clear empirical gains, and ability to enable early stopping with significant efficiency improvements. Concerns were mainly around limited related work discussion, questions about generalization and memory overhead, and some presentation clarity; however, these were mostly addressed in the rebuttal with additional experiments and improved explanations. The authors should also take reviewers' feedback and their responses during the rebuttal to revise the paper accordingly.

Overall, the paper provides a novel and practically impactful perspective on leveraging internal model signals for decoding, and is likely to stimulate further work in this direction, leading to a clear acceptance.